# Visual Cues for Turning in Parkinson’s Disease

**DOI:** 10.3390/s22186746

**Published:** 2022-09-07

**Authors:** Julia Das, Rodrigo Vitorio, Allissa Butterfield, Rosie Morris, Lisa Graham, Gill Barry, Claire McDonald, Richard Walker, Martina Mancini, Samuel Stuart

**Affiliations:** 1Department of Sport, Exercise and Rehabilitation, Northumbria University, Newcastle NE1 8ST, UK; 2Northumbria Healthcare NHS Foundation Trust, North Shields NE29 8NH, UK; 3Gateshead Health NHS Foundation Trust, Gateshead NE8 2PJ, UK; 4Department of Neurology, Oregon Health & Science University, Portland, OR 97239, USA

**Keywords:** Parkinson’s disease, turning, visual cues, wearable sensor, mobility

## Abstract

Turning is a common impairment of mobility in people with Parkinson’s disease (PD), which increases freezing of gait (FoG) episodes and has implications for falls risk. Visual cues have been shown to improve general gait characteristics in PD. However, the effects of visual cues on turning deficits in PD remains unclear. We aimed to (i) compare the response of turning performance while walking (180° and 360° turns) to visual cues in people with PD with and without FoG; and (ii) examine the relationship between FoG severity and response to visual cues during turning. This exploratory interventional study measured turning while walking in 43 participants with PD (22 with self-reported FoG) and 20 controls using an inertial sensor placed at the fifth lumbar vertebrae region. Participants walked straight and performed 180° and 360° turns midway through a 10 m walk, which was done with and without visual cues (starred pattern). The turn duration and velocity response to visual cues were assessed using linear mixed effects models. People with FoG turned slower and longer than people with PD without FoG and controls (group effect: *p* < 0.001). Visual cues reduced the velocity of turning 180° across all groups and reduced the velocity of turning 360° in people with PD without FoG and controls. FoG severity was not significantly associated with response to visual cues during turning. Findings suggest that visual cueing can modify turning during walking in PD, with response influenced by FoG status and turn amplitude. Slower turning in response to visual cueing may indicate a more cautious and/or attention-driven turning pattern. This study contributes to our understanding of the influence that cues can have on turning performance in PD, particularly in freezers, and will aid in their therapeutic application.

## 1. Introduction

Parkinson’s disease (PD) is a progressive neurological disorder that is associated with a lack of dopamine producing cells within the basal ganglia of the brain [1]. The main symptoms associated with PD are tremor, bradykinesia, akinesia, rigidity of muscles and postural instability [2]. Freezing of gait (FoG) is a common symptom of the later stages of PD, often described as a sudden inability to initiate or continue a gait cycle [3]. FoG manifests itself as trembling legs, small shuffling steps, or no movement of the limbs at all. Although this symptom is commonly experienced when a person is turning or trying to initiate walking, it can also be seen when faced with a lack of space, obstacles, a stressful situation, or distraction [4].

FoG is often accompanied by a change in stride length, leading to shorter steps, particularly in the run up to an event such as a turn; this can lead to a further complication known as a turning deficit [5]. People with PD often turn in a way that is described as “enbloc”, which is almost simultaneous turning of the different segments of the body, rather than the rotations occurring independently between the segments. Alongside this impairment, the typical turn of a person with PD is often slower, consists of more steps than the average person and involves a narrower base of support (BoS), a larger turning arc, a potential forward lean and instability of the person’s centre of mass (CoM) [6]. An average of 45–68% of people with PD fall each year, and a large proportion of these fall recurrently [7]. Falls are associated with higher rates of hospital admission, injury prevalence and are a large contributor to loss of independence, increase in disability and reduced quality of life (QoL) in people with PD [7,8].

Although PD is incurable, there are various ways of treating its symptoms. Antiparkinsonian medications, such as levodopa, are used to manage PD symptoms [9]. This has been shown to improve arm swing range, gait velocity and stride length but can also cause an increase in postural sway, turning speed and arm swing velocity [10]. The use of medications has been unsatisfactory within PD symptom treatment due to the refractory nature of some symptoms in relation to dopamine [10]. Levodopa has only been seen to improve gait measures related to pace, and thus cues are commonly used in physiotherapy practice to help alleviate the deficits and symptoms that medications do not aid [11].

Cueing is a useful technique used by physiotherapists alongside medications, as it can help focus an individual’s attention on their gait [12,13,14]. Cues can be defined as temporal or spatial stimuli which can be used as triggers to initiate a movement, and are usually provided as visual, auditory or tactile signals [12]. They aim to prevent FoG which in turn can preserve functional gait. Cues can also serve as a rescue strategy if a person does find themselves in a freezing episode [12]. There is limited evidence regarding turning and cueing within PD, particularly with visual cues, thus leaving it unclear how useful cues are for turning in clinical practice.

Given the limited research focusing on visual cues as an intervention for turning deficit in PD, this study aims to: (1) compare the response of turning performance while walking (180° and 360° turns) to visual cues in people with PD with (PD+FoG) and without FoG (PD-FoG); and (2) examine the relationship between FoG severity and response to visual cues during turning. We hypothesise that turning will be improved (i.e., increased velocity and reduced duration) with visual cues, particularly in those who self-report FoG. Additionally, we believe that response to visual cues will depend on FoG severity, with those that have more severe FoG benefitting the most from visual cues.

## 2. Methodology

### 2.1. Participants

This study is an exploratory interventional study of the response to visual cues during turns while walking in people with PD who do and do not report FoG. A total of 43 PD participants (n = 22 with self-reported FoG and n = 21 without FoG) were recruited by neurologists at the Oregon Health and Science University (OHSU) Movement Disorders Clinic. Self-reported FoG was based upon the Freezing of Gait Questionnaire (FOGQ) [15]. Subjects were categorized as “freezers” if they had experienced such a feeling or episode within the month prior. This study was ethically approved by an Oregon Health and Science University (OHSU) Institutional Review Board (#9903). All subjects provided their written informed consent prior to the experiment prior to involvement.

### 2.2. Inclusion/Exclusion Criteria

*Inclusion criteria:* Clinical diagnosis of PD by a movement disorder specialist according to UK brain bank criteria, Hoehn and Yahr stage II–III [16], aged ≥ 50 years, adequate vision and hearing (Snellen chart visual acuity ≥ 12/18), and able to walk and stand unaided.

*Exclusion criteria:* Cognitive impairment (score: MoCA < 21 and <10 CLOX1), unstable medication for one month prior to study, psychiatric co-morbidity, acute lower back or lower/upper extremity pain, peripheral neuropathy, unable to comply with protocol and rheumatic and orthopaedic diseases affecting balance and gait.

### 2.3. Data Collection

Data were collected at the Balance Disorders Laboratory, Oregon Health and Science University, Portland, OR, USA. Each participant attended a 2-h session at the laboratory. Participants underwent a battery of demographic, clinical and cognitive assessments. The following tests were administered: the Movement Disorder Society Unified Parkinson’s disease Rating Scale Motor Subscale (MDS-UPDRS III) [17]. *Global cognition* was assessed with the Montreal Cognitive Assessment (MoCA) [18]. *Attention* was measured with a computerized button pressing battery, involving simple reaction time (SRT), choice reaction time (CRT) and digit vigilance. *Executive function* was measured using Royall’s clock drawing (CLOX 1&2) [19] and Trail Making Part B-A [20]. Working memory and visuo-spatial ability was measured through seated forward digit span and judgement of line orientation (JLO) tasks [21], respectively. Basic visual functions of visual acuity and contrast sensitivity were assessed using standardised charts (logMar and logCS).

Participants performed two turning tasks under two different conditions with, and without, visual cues, in their usual ON medication state (within 60 min of taking anti-Parkinsonian medications). Turning tasks included walking 10m with turns of 180° and 360° towards their non-dominant side at the half-way point of the walk. The visual cue involved taped black lines on the floor in a starred pattern (Figure 1), which participants were asked to step over when turning (e.g., four two-inch wide black tape lines that extended ~1m from the centre of the star, providing 8 points). Participants wore nine Opal (Version 2, APDM, Inc., Portland, OR. USA) inertial measurement units (IMUs) on both feet, calves, wrists, lumbar region, sternum and head during all tasks (Figure 1) [22].

### 2.4. Data Analysis

Raw IMU data from the IMU on the lumbar region were analysed with validated custom-made MATLAB algorithms to derive turning characteristics [23,24]. Turning was detected by the horizontal rate of the sensor placed on the participant’s lumbar spine. For a movement to be defined as a turn, the trunk had to rotate around a horizontal axis at least 45°, accompanied by at least one step of each foot. For a turn to be considered, it had to last between 0.5 and 10 s. Integration of the angular rate of the lumbar sensor about the vertical axis was used to calculate the relative turn angle. The primary outcome measure for this study is turning performance, specifically turn velocity (i.e., the peak angular velocity of the turn) and duration (i.e., time from turn onset to end). Secondary outcomes include clinical measures and cognitive and visual function assessments.

### 2.5. Statistical Analysis

Statistical analysis was performed with SPSS version 26 (IBM, Armonk, NY, USA). Normality of data was checked with Kolomogrov-Smirnov tests and visual inspection of box-plots. One-way analysis of variance (ANOVA) was used to compare continuous demographic variables across all groups (controls, PD-FoG and PD+FoG), and *t*-tests were used to compare individual group comparisons (controls vs. PD-FoG, controls vs. PD+FoG and PD-FoG vs. PD+FoG), and chi-square analysis for ordinal data. Separate linear mixed effects models (LMEMs) were conducted for each turning metric (duration and velocity) to examine the effect of the intervention (Cue and No Cue), with group (controls, PD-FoG and PD+FoG) as a between subjects factor. Post hoc tests were used to locate differences where applicable (with adjustments for multiple comparisons: 0.05/number of comparisons). Additionally, to better isolate the influence of FoG status itself rather than disease severity on the response to cueing, similar LMEMs were conducted while controlling for MDS-UPDRS-III score (group: PD-FoG and PD+FoG; condition: Cue and No Cue). Pearson’s correlations were conducted to examine relationship between change scores (Cue and No Cue) of turning and FoG severity (represented by the new FOGQ). The significance level was set to *p* < 0.05.

## 3. Results

### 3.1. Participant Characteristics

The participant’s demographic, visual, cognitive and clinical features are summarised within Table 1. The representative groups for PD (both PD+FoG and PD-FoG) and healthy controls were well matched in terms of age, sex, height, education, visual ability and global cognition. No significant differences in demographic, visual or cognitive variables were observed between PD-FoG and healthy controls. PD+FoG had significantly worse disease severity (MDS-UPDRS III), more advanced disease stage (H&Y stage), longer disease duration, greater levodopa medication dosage, slower CRT and more falls in the previous 12 months than PD-FoG. Additionally, PD+FoG had significantly greater levels of depressive symptoms and slower/poorer performance in the Trail Making test (parts A and B) and reaction time tests.

### 3.2. Turning Performance without and with Visual Cues

PD+FoG turned longer and slower (180° and 360°) than PD-FoG and healthy controls (Table 2 and Figure 2), with group main effects observed for all turn outcomes, even when controlling for disease severity (Table 3). Post hoc tests revealed significant differences between PD+FoG and PD-FoG or healthy controls for all turn outcomes, but no significant differences were observed between PD-FoG and healthy controls (Table 2, Figure 2).

Interestingly, the use of visual cues reduced velocity of turning 360° only in healthy individuals and PD-FoG (Table 2 and Table 3). A significant group X condition interaction was observed for velocity of turning 360° in both LMEMs. Post hoc tests revealed that while healthy individuals and PD-FoG had reduced turn velocity in response to visual cues, PD+FoG did not significantly change velocity of turning 360° across turning conditions (Table 2 and Table 3). Further, a significant main effect of condition revealed that the use of visual cues reduced velocity of turning 180° in all groups (Table 2 and Table 3).

No significant condition main effects or group X condition interactions were observed for turn duration (180° or 360°, Table 2 and Table 3) and correlational analysis showed that FoG severity was not significantly associated with response to visual cues during turning (*p* > 0.05, Appendix A Table A1).

## 4. Discussion

This study investigated turning performance when walking in response to visual cueing in people with PD with and without FoG. Our results showed that turn velocity significantly reduced with the use of a visual cue, compared to without, in all groups during 180° turns. Alternatively, turn velocity only significantly reduced in healthy controls and PD-FoG during 360° turns, but not in PD+FoG. Interestingly, the response to visual cues during turning was not associated with FoG severity. These results suggest that visual cueing can modify turning during walking in PD, with response influenced by FoG status and turn amplitude.

### 4.1. Turning Performance in Parkinson’s Disease

Without cueing, turning was poorer in PD+FoG compared to both PD-FoG and healthy controls, with reduced velocity and increased duration during both 180° and 360° turns. Interestingly, we found no significant differences in turning between PD-FoG and healthy controls for velocity or duration of a turn (180° and 360°). Our results suggest that turning 180° or 360° when walking may be impaired in those with FoG, which is not surprising based on the previous literature of turning deficit in PD+FoG compared to PD-FoG within laboratory and free-living assessments. However, the lack of significant deficit in turning performance for PD-FoG compared to healthy controls is contradictory to some of the previous literature [25]; but may highlight the lack of sufficient FoG classification in other studies (i.e., previous turning studies have not consistently reported whether FoG was examined or self-reported in their cohorts [26]).

### 4.2. Turn Response to Visual Cues

Our findings showed that turning performance, velocity and duration during walking was not improved with the use of visual cues in people with PD or healthy controls. In fact, visual cues slowed turn performance in healthy controls and PD-FoG, and did not change turn performance in PD+FoG. Our findings support and contradict previous research of various cueing modalities for turning in PD [27,28,29,30,31,32]. For example, Mancini, Smulders, Harker, Stuart and Nutt [29], Spildooren, Vercruysse, Meyns, Vandenbossche, Heremans, Desloovere, Vandenberghe and Nieuwboer [30] and Willems, Nieuwboer, Chavret, Desloovere, Dom, Rochester, Kwakkel, Van Wegen and Jones [32] observed slower or poorer turn performance (i.e., more steps in a turn, slower turns, wider turning arc, etc.) in response to auditory or tactile cueing in people with PD. Our previous findings have also shown slower turns when using open- (metronome-like) or closed-loop (synchronised to individuals’ step time) tactile cueing in people with PD with and without FoG [27]. Additionally, visual cues delivered with transverse tape lines on the floor, or by augmented reality, have both been shown not to change FOG occurrence and slowed turn performance in PD [33]. In contrast, Gómez-González, Martín-Casas and Cano-de-la-Cuerda [28] examined the impact of auditory cues on turn performance in PD, showing that auditory cues increased turn cadence. In the largest study of cueing for turning in PD to date (n = 133), Nieuwboer, Baker, Willems, Jones, Spildooren, Lim, Kwakkel, Van Wegen and Rochester [31] studied the impact of auditory, visual and tactile cueing on turn performance in PD-FoG and PD+FoG, showing that all cues increased the speed of turning, but auditory cues significantly increased turning speed more than visual cues (i.e., a flashing LED light on a pair of glasses, synchronised to the person’s step frequency).

Unlike gait, it is unclear what the clinical aim of using cueing modalities for turning in PD should be, i.e., whether cueing should aim to increase or decrease turning speed. The current study showed a reduction in turning speed that may be due to participants taking a more cautious approach to turning. Indeed, Mancini, Smulders, Harker, Stuart and Nutt [29] previously found that tactile cueing slowed turns, but improved the smoothness (quality) of turning in people with PD, which aligns with the current study. In addition, improved cranio-caudal control during walking turns in PD (i.e., from “enblock” to head first strategy) has been reported after 10 trials of practice with visual cues [34]. Additionally, Mellone, Mancini, King, Horak and Chiari [6] previously reported that asking individuals with PD to turn quickly increased their instability, with recommendations that a wider turning arc and slower turn speed may reduce the risk of falling when turning in PD. It is believed that cueing may slow turn speed due to requiring increased attention to concentrate on the task [27], which is similar to visual cue response during gait being underpinned by attentional and visual-attentional processing [35,36]. Therefore, findings of the current study suggest that reduced turn speed with visual cues may be beneficial in PD, particularly for reducing falls risk.

There are differences in the turning protocols (cue modality and turning task) and delivery of cueing, particularly visual cueing, between studies that may influence the turning response in PD. Despite some studies showing the benefit of particular cueing methods over visual cues [31], there remains no ‘gold-standard’ method for cue delivery in terms of modality (auditory, tactile or visual) and implementation for people with PD. Previous studies have examined turning performance when walking or when turning-in-place; and have delivered visual cues using traditional methods (i.e., tape on the floor) or modern technology (i.e., LED glasses, augmented reality. etc.), with differences in methodologies impacting on results and comparison to the current study. Until a standardised or personalised visual cue implementation and assessment methodology for turning is developed, comparisons between study outcomes need to be made with caution, as subtle differences in the delivery of visual cues may alter findings. For example, although taped lines and augmented reality visual cues have been reported to have a similar influence on turning in PD [33], the use of augmented reality headsets has been said to potentially add to turning burden due to the headset device involved being a distraction of attention [37]. Ultimately, the lack of clear understanding of the mechanisms underlying visual cues for turning in PD limits the development of more effective and standardised methodologies.

### 4.3. Clinical Implications

Visual cues slow turn performance and this could, in turn, lower falls risk for people with PD. This suggests that clinical application of visual cues should entail education about the speed that the individual turns when using the cues. For example, people with PD could be advised that the visual cues will slow their turning performance but that they aim to enhance focus on turning to avoid instability and, ultimately, falls. However, further research is required around the best kind of visual cue to facilitate turning deficit and to explore the implications of using a visual cue long term.

### 4.4. Limitations

There are several limitations to consider. First, the sample size of the groups is modest so caution must be taken when extrapolating the results from this study. Second, our findings refer to the immediate effects (single exposure) of visual cues on turning, which may reflect response seen within clinic settings, but long-term effects (habitual effects) of cueing at home are still to be investigated.

## 5. Conclusions

This study found that providing a visual cue may reduce turn speed in older adults and those with PD, which may indicate a more cautious turning pattern with visual cues. Findings correspond to immediate use of traditional tape visual cues during walking, therefore future work is required to understand response to different visual cue modalities (i.e., new technology), as well as use of visual cues over prolonged periods. Future studies are also required to understand the underlying mechanisms involved in the response not just to visual cues, but also to other modalities of cueing (such as auditory and/or tactile cues) during turning in PD.

## Figures and Tables

**Figure 1 sensors-22-06746-f001:**
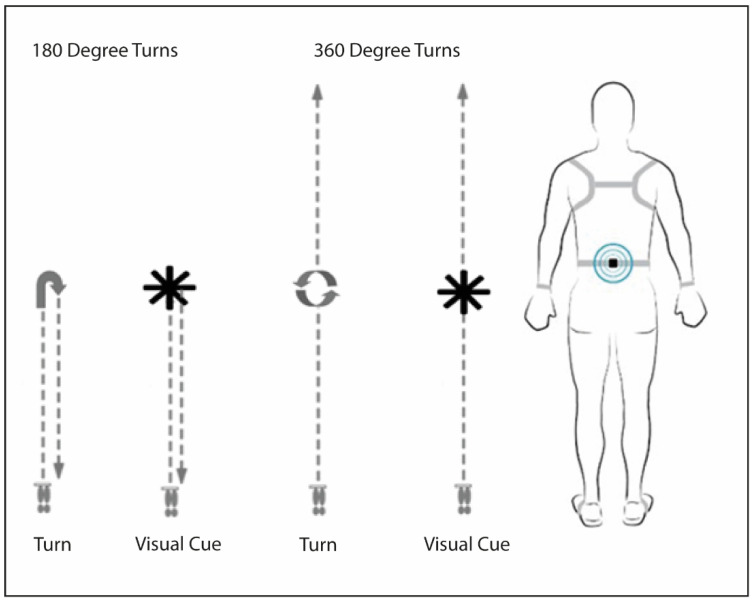
Turning conditions with and without visual cue, and IMU placement.

**Figure 2 sensors-22-06746-f002:**
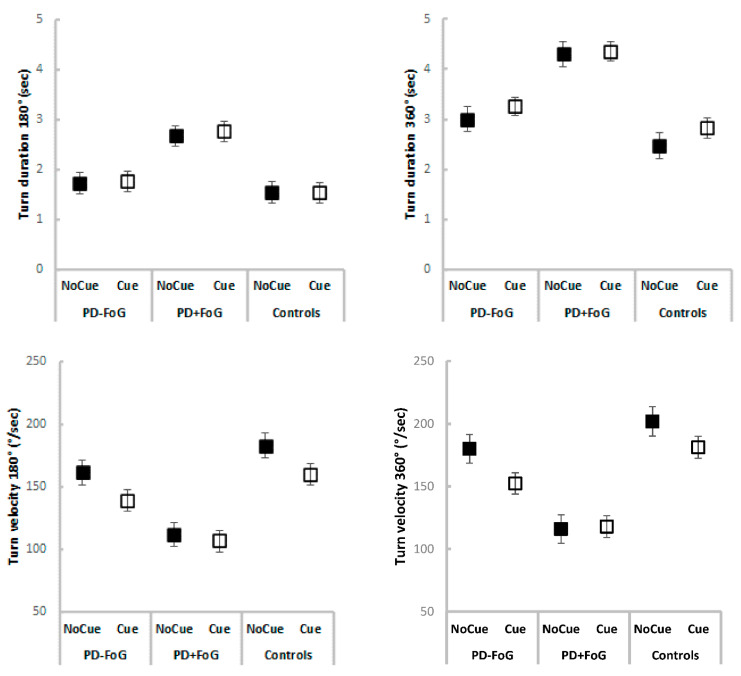
Turn outcomes (means and standard errors) for each group and cue condition.

**Table 1 sensors-22-06746-t001:** Participant demographics.

Variables	PD-FoG (n = 21) Mean (SD)	PD+FoG (n = 22) Mean (SD)	Controls (n = 20) Mean (SD)	Overall *p*	PD-FoG vs. PD+FoG *p*	PD-FoG vs. Controls *p*	PD+FoG vs. Controls *p*
*Age (years)*	69.3 (5.0)	68.5 (7.9)	69.9 (8.2)	0.816	0.682	0.792	0.576
*Sex*	11M/10F	15M/7F	10M/10F	0.426	0.170	0.123	0.639
*Height (cm)*	169.6 (11.4)	174.8 (9.3)	168.8 (10.0)	0.124	0.108	0.815	0.050
*Weight (kg)*	76.9 (19.5)	86.1 (17.0)	72.6 (14.3)	**0.039 ***	0.105	0.435	**0.009 ***
*Education (years)*	17.1 (2.8)	16.7 (2.5)	17.5 (2.5)	0.560	0.573	0.623	0.747
*Depression scale (GDS-15)*	4.9 (5.1)	7.3 (4.2)	3.3 (4.2)	**0.020 ***	0.105	0.263	**0.004 ***
*Retrospective falls (last 12 months)*	0.33 (0.58)	5.91 (11.55)	0.50 (1.10)	**0.014 ***	**0.034 ***	0.544	**0.040 ***
*Visual acuity (binoc)*	0.06 (0.18)	0.08 (0.11)	0.05 (0.12)	0.727	0.628	0.805	0.344
*Contrast sensi-tivity (binoc)*	1.57 (0.20)	1.56 (0.14)	1.65 (0.13)	0.150	0.828	0.145	**0.034 ***
*Judgement of line orientation*	24.8 (4.7)	22.8 (8.1)	26.4 (3.6)	0.157	0.323	0.251	0.071
*Montreal cognitive assessment*	27.6 (2.2)	26.9 (2.7)	27.0 (2.1)	0.559	0.323	0.407	0.806
*Digit span*	6.3 (1.2)	5.8 (0.9)	6.0 (1.0)	0.273	0.121	0.302	0.429
*CLOX 1*	12.9 (1.7)	12.5 (1.6)	13.3 (1.4)	0.366	0.544	0.421	0.150
*CLOX 2*	13.9 (1.2)	13.9 (0.8)	13.5 (1.5)	0.391	0.894	0.278	0.269
*Trail Making Test A (s)*	30.8 (16.7)	37.7 (17.5)	22.6 (6.4)	**0.007 ***	0.193	0.052	**0.001 ***
*Trail Making Test B (s)*	67.0 (36.4)	97.2 (68.0)	53.4 (17.0)	**0.012 ***	0.079	0.136	**0.008 ***
*Trail Making Test B-A (s)*	36.1 (27.1)	54.9 (61.0)	30.8 (14.1)	0.136	0.204	0.430	0.084
*Simple reaction time (ms)*	351.1 (66.2)	397.8 (84.6)	333.8 (35.8)	**0.007 ***	0.051	0.302	**0.003 ***
*Choice reaction time (ms)*	519.6 (97.5)	619.1 (173.5)	500.0 (65.3)	**0.005 ***	**0.023 ***	0.458	**0.006 ***
*Disease duration (years)*	5.5 (2.9)	9.6 (7.0)	¯	¯	**0.017 ***	¯	¯
*H&Y scale (stage)*	II(21)/III(0)	II(18)/III(4)	¯	¯	**0.042 ***	¯	¯
*LEDD (mg/day)*	628.6 (373.8)	971.5 (396.6)	¯	¯	**0.006 ***	¯	¯
*MDS-UPDRS III*	25.4 (12.3)	38.5 (13.3)	¯	¯	**0.002 ***	¯	¯
*FOGQ*	0 (0)	15.4 (7.7)	¯	¯	**<0.001 ***	¯	¯

* Statistically significant (*p* < 0.05, emboldened).

**Table 2 sensors-22-06746-t002:** Results from the liner mixed effects models (condition: Cue, No Cue; group: controls, PD-FoG and, PD+FoG) for turn outcomes.

Outcomes	Group F (*p*) (Post Hoc)	Condition F (*p*) (Post Hoc)	Group × Condition F (*p*) (Post Hoc)
** *Turn duration* **			
180°	**10.777 (<0.001 *) PD+FoG > HC, PD-FoG**	0.167 (0.684)	0.090 (0.914)
360°	**18.145 (<0.001 *) PD+FoG > HC, PD-FoG**	3.287 (0.077)	0.558 (0.576)
** *Turn velocity* **			
180°	**14.772 (<0.001 *) PD+FoG < HC, PD-FoG**	**2.508 (0.001 *)** **Cue < NoCue**	1.484 (0.236)
360°	**16.359 (<0.001 *) PD+FoG < HC, PD-FoG**	**11.063 (0.002 *)** **Cue < NoCue**	**3.607 (0.034 *)** **HC, PD-FoG: Cue < NoCue** **PD+FoG: Cue = NoCue**

[HC: healthy controls, PD+FoG: Parkinson’s disease with freezing of gait, PD-FoG: Parkinson’s disease without freezing of gait * Statistically significant (*p* < 0.05, emboldened).

**Table 3 sensors-22-06746-t003:** Turn outcomes for PD groups (PD-FoG vs. PD+FoG) and condition, with results from the liner mixed effects models while controlling for disease severity (UPDRS-III).

Outcomes	PD-FoG	PD+FoG	Group	Condition	Group × Condition
	Mean	Mean	F (*p)*	F (*p*)	F (*p*)
	(SE)	(SE)			(Post Hoc)
** *Turn duration (s)* **					
180° NoCue	1.81	2.57			
	(0.09)	(0.31)	**7.738 (0.009 *)**	0.124 (0.728)	0.009 (0.927)
180° Cue	1.85	2.64	**PD+FoG > PD-FoG**		
	(0.08)	(0.29)			
360° NoCue	3.16	4.08			
	(0.22)	(0.40)	**6.064; 0.02 ***	1.398 (0.249)	0.228 (0.637)
360° Cue	3.41	4.19	**PD+FoG > PD-FoG**		
	(0.16)	(0.28)			
** *Turn velocity (°/s)* **					
180° NoCue	154.2	119.8			
	(7.1)	(9.7)	**7.129 (0.011 *)**	**6.442 (0.015 *)**	2.461 (0.125)
180° Cue	131.8	114.5	**PD+FoG < PD-FoG**	**Cue < NoCue**	
	(7.6)	(5.4)			
360° NoCue	171.8	126.8			**6.382 (0.016 *)**
	(8.9)	(9.9)	**7.692 (0.008 *)**	**4.906 (0.033 *)**	**PD-FoG: Cue < NoCue**
360° Cue	144.1	128.7	**PD+FoG < PD-FoG**	**Cue < NoCue**	**PD+FoG: Cue = NoCue**
	(7.8)	(6.9)			

[SE: Standard error of the mean, PD+FoG: Parkinson’s disease with freezing of gait, PD-FoG: Parkinson’s disease without freezing of gait]. * Statistically significant (*p* < 0.05, emboldened).

## Data Availability

The data presented in this study are available on request from the corresponding author.

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
