# Peer review of "Visual Cues for Turning in Parkinson’s Disease"

_sensors, 2022, doi:10.3390/s22186746_

Round 1

Reviewer 1 Report

Visual Cues for Turning in Parkinson’s Disease studied whether the visual cue may help the PD patients w/ FoG to improve their gaits. They found that their visual cue did not help PD patients w/ FoG to improve their gaits.

Not only for PD patients w/ FoG, but also for those PD patients with normal gaits and those normal people, visual cue did not improve their performance. The idea of visual cue is helping the individual focusing on their gaits. There is no reason that visual cue even blocks the gait performance of those people w/ normal gaits. However, if the visual cue decreased the gait performance of those normal people, the visual cue should absolutely decrease the gait performance of those PD patients w/ FoG. Please describe the visual cue in this research or show it with an image. Please explain how the visual cue in this research improves the gaits performance. Is it possible to choose a good visual cue which can improve the gait performance for this research? 

Author Response

Please see attachment for cover letter with table of responses to all reviewer comments.

We thank the Reviewer for their time and efforts in reviewing our manuscript. We have made changes in the manuscript highlighted in yellow and provide detailed responses to comments in the tables below.

Reviewer #1

Reviewer Comment

Response

Visual Cues for Turning in Parkinson’s Disease studied whether the visual cue may help the PD patients w/ FoG to improve their gaits. They found that their visual cue did not help PD patients w/ FoG to improve their gaits.

Not only for PD patients w/ FoG, but also for those PD patients with normal gaits and those normal people, visual cue did not improve their performance. The idea of visual cue is helping the individual focusing on their gaits. There is no reason that visual cue even blocks the gait performance of those people w/ normal gaits. However, if the visual cue decreased the gait performance of those normal people, the visual cue should absolutely decrease the gait performance of those PD patients w/ FoG.

Please describe the visual cue in this research or show it with an image.

Please explain how the visual cue in this research improves the gaits performance.

Is it possible to choose a good visual cue which can improve the gait performance for this research?

The visual cue used is depicted in Figure 1. It is a commonly used clinical intervention for turning impairment in Parkinson’s, which involves a starred pattern on the floor that patients step over the lines of when they are on top of it.

We used this pattern of visual cue as it is commonly used in clinical practice, but there could be other patterns that may improve turning that were beyond the scope of this study to determine.

The visual cue (shown in Figure 1) that was used is meant to improve turning by increasing patient attention to the task and getting them to step over lines on the floor, which may improve turn metrics through guiding foot placement and encourage lifting of the feet. However, our results show that the visual cue slows turning in Parkinson’s, which may actually be a positive finding, as patients are unstable when turning (even at their usual, slow speed) and therefore increasing their speed may increase risk of falls. It is likely that the patients took a cautious approach to turning with the visual cue, which means they were slower but likely more stable. This is an interesting finding which we have focused on within the discussion of the manuscript.

Reviewer 2 Report

In this paper, the authors have studied visual cues that have been shown to improve general gait characteristics in Parkinson’s disease (PD). There are several main concerns are need to be addressed before I can recommend this paper for a publication.  

1.The authors should reorganize the Introduction part as follows: background, limit of prior arts, research motivation, main contribution, which may lead to a clear logic about why this work is motivated. 

2. In particular, the authors should summarize the main contribution in the Abstract part.

3.More graphs should be used to help potential readers to understand tables. There are too many tables.

4. Section 4.4 should be merged into the conclusion part.

5. The conclusion part should discuss how this work motivates other researchers in this area.

6. I also recommend the authors to refer to several recent related literatures in the introduction part.

a. 10.3390/s20051510

b. 10.1109/JSEN.2022.3149337

Author Response

We would like to thank the Reviewer for their time and efforts in reviewing our manuscript. We have made changes in the manuscript highlighted in yellow and provide detailed responses to comments in the table below:

Reviewer Comment

Response

In this paper, the authors have studied visual cues that have been shown to improve general gait characteristics in Parkinson’s disease (PD). There are several main concerns are need to be addressed before I can recommend this paper for a publication.  

1.The authors should reorganize the Introduction part as follows: background, limit of prior arts, research motivation, main contribution, which may lead to a clear logic about why this work is motivated. 

The introduction is currently laid out in a similar manner to the reviewer’s suggestion. Providing a background on Parkinson’s disease, details on the gait / turning deficits in the condition, rationale for treatment with visual cueing in clinical practice, and finally clear aims for the study. There is limited evidence regarding turning and cueing within PD, particularly with visual cues, thus leaving it unclear how useful cues are for turning in clinical practice.

2. In particular, the authors should summarize the main contribution in the Abstract part.

The main contribution was summarised in the abstract. Specifically, visual cues have not been examined for improving turning in Parkinson’s and therefore we examined the effect, with results showing that they slowed turning (which is likely an indicator of a more cautious approach to turning being taken by the patients). We have clarified this in the abstract:

“Slower turning in response to visual cueing may indicate a more cautious and/or attention-driven turning pattern. This study contributes to our understanding of the influence that cues can have on turning performance in PD, particularly in freezers, and will aid in their therapeutic application.”

3.More graphs should be used to help potential readers to understand tables. There are too many tables.

We have provided three tables, with one showing the demographics of the participants. We have included an additional figure (Figure 2) as a visual representation of the means/SE previously shown in Table 2. To avoid double presentation of data, we have deleted the means/SE from Table 2 (but have kept the statistical results in Table 2)

4. Section 4.4 should be merged into the conclusion part.

To ensure clarity of the limitations of the study to the reader, we have chosen to keep the limitations section separate from the conclusions.

5. The conclusion part should discuss how this work motivates other researchers in this area.

The conclusions provide several sentences on the motivation and direction for future work in this area, specifically;

“Findings correspond to immediate use of traditional tape visual cues during walking, therefore future work is required to understand response to different visual cue modalities (i.e., new technology), as well as use of visual cues over prolonged periods. Future studies are also required to understand the underlying mechanisms involved in the response not just to visual cues, but also to other modalities of cueing (such as auditory and/or tactile cues) during turning in PD.”

6. I also recommend the authors to refer to several recent related literatures in the introduction part.

a. 10.3390/s20051510

b. 10.1109/JSEN.2022.3149337

Thank you for the suggestions, however these studies are not relevant for the current manuscript. One involves skin mounted sensors detecting lumbar-pelvic movement and the other involves an algorithm to detect human movement activity, both of which are in healthy individuals. The current study involves use of a belt worn inertial sensor to measure turning when walking in a laboratory in response to a clinical visual cue intervention in those with Parkinson’s disease. Therefore, we have not included these references in our introduction.

Reviewer 3 Report

This article mentions the speed of 180 degree and 360 degree turns, but I didn't know what kind of data was used to judge the speed.

Add the coordinate system to the IMU placement in Figure.1.

How was the angular velocity integrated?

It is also known that errors accumulate with the time it takes to integrate the angular velocity.

Have you taken measures against the error?

..

The t-test is used to compare each group.

However, when multiple comparisons of groups are performed by t-test, the null hypothesis may be rejected even though it should not be rejected.

You should use the Multiple Comparison Procedure.

How was Turn duration and Turn Velocity measured (calculated)?

"FoG severity was not associated with response to visual cues during turning (p> 0.05, 179 Supplemental table 1)."

"Interestingly, we found no significant differences in turning between PD-FOG and 200 healthy controls for velocity or duration of a turn (180, 360)."

This expression is misleading.

Even if the null hypothesis is rejected, it cannot be proved to be irrelevant.

The t-test just couldn't find a significant difference.

The major problems in this article are the following three points.

1. There is a possibility that false positives are high without multiple comparisons.

2. The contribution of this article is based on the fact that the hypothesis could not be rejected and makes a logic.

3.  I didn't quite understand the rationale that slower turning speeds make it harder to fall.

Isn't it natural for PD patients with FoG to have slow turning speeds?

Also, when checking the visual stimulus, it is a normal human reaction that the turning speed slows down, and the result is predictable.

Extensive analysis is required using EEG and gaze measurement data to confirm the effect of visual stimuli.

Therefore, I consider the novelty and contribution of this article to be small.

It will be necessary to review the method of hypothesis testing and the method of interpretation.

Author Response

We thank the Reviewer for their time and efforts in reviewing our manuscript. We have made changes in the manuscript highlighted in yellow and provide detailed responses to comments in the tables below:

Reviewer Comment

Response

This article mentions the speed of 180 degree and 360 degree turns, but I didn't know what kind of data was used to judge the speed.

Add the coordinate system to the IMU placement in Figure.1.

How was the angular velocity integrated?

It is also known that errors accumulate with the time it takes to integrate the angular velocity.

Have you taken measures against the error?

How was Turn duration and Turn Velocity measured (calculated)?

This article does not detail the turning algorithm used to obtain the turn outcomes, as this was not the aim of the study. Instead, we refer to our previously published and validated turning algorithm work in Parkinson’s disease, which has been integrated into the Mobilitylab wearable sensor system that was used in our testing. We briefly describe how turn was analysed in Section 2.4. For further details about the algorithm, please see references below:

Mancini, M.; Schlueter, H.; El-Gohary, M.; Mattek, N.; Duncan, C.; Kaye, J.; Horak, F. B., Continuous monitoring of turning mobility and its association to falls and cognitive function: a pilot study. Journals of Gerontology Series A: Biomedical Sciences and Medical Sciences 2016, 71, (8), 1102-1108.

El-Gohary, M.; Pearson, S.; McNames, J.; Mancini, M.; Horak, F.; Mellone, S.; Chiari, L., Continuous monitoring of turning in patients with movement disability. Sensors 2013, 14, (1), 356-369.

We have added short definitions of turn duration and velocity within Section 4.2 to aid clarity.

"FoG severity was not associated with response to visual cues during turning (p> 0.05, 179 Supplemental table 1)."

"Interestingly, we found no significant differences in turning between PD-FOG and 200 healthy controls for velocity or duration of a turn (180, 360)."

This expression is misleading.

Even if the null hypothesis is rejected, it cannot be proved to be irrelevant.

The t-test just couldn't find a significant difference.

We understand that the reviewer is suggesting that despite a non-significant result, there may be differences in the turning data. However, without a significant result we cannot be sure that differences existed. However, we have included the term ‘no significant differences’, rather than ‘no differences’ were found.

We have changed the association sentence to align with this:

“FoG severity was not significantly associated with response to visual cues during turning (p> 0.05, 179 Supplemental table 1)."

We acknowledge that larger sample sizes may reveal a significant result and have referenced sample size as a potential limitation in section 4.4.

The t-test is used to compare each group.

However, when multiple comparisons of groups are performed by t-test, the null hypothesis may be rejected even though it should not be rejected.

You should use the Multiple Comparison Procedure.

The major problems in this article are the following three points.

1. There is a possibility that false positives are high without multiple comparisons.

The statistical analysis details that t-tests were only used to compare demographic data for individual groups (e.g., PD-FOG vs PD+FOG) as a post-hoc test following the one-way ANOVA used to examine overall group (HC, PD-FOG, PD+FOG) differences.

We did not control for multiple comparisons in the statistical analysis of demographics as these data were not the primary data for the study (e.g., turning data).

Linear mixed effects models were used to compare the turning data (velocity and duration) from the participants (HC, PD-FOG, PD+FOG). Post hoc tests used to locate differences applied the adjustment of significance level (0.05 / number of comparisons). To make this point clear in the manuscript, we have added the following sentence to section 2.5: “Post hoc tests were used to locate differences where applicable (with adjustments for multiple comparisons: 0.05 / number of comparisons).”

2. The contribution of this article is based on the fact that the hypothesis could not be rejected and makes a logic.

We hypothesised that turning velocity would increase and duration decrease in response to visual cueing in those with Parkinson’s, particularly those with freezing of gait. However, our results show that visual cueing significantly slowed turn velocity and increased duration, which is the opposite of our hypothesis. However, this may indicate that people with Parkinson’s took a more cautious approach to turning, which may be safer for them. This is due to people with Parkinson’s struggling to turn safely / steadily during their usual mobility, and therefore increasing speed of turns may cause further instability, so visual cues may improve task performance by reducing speed and decreasing the risk of falls during turns. In line with this interpretation, Mellone et al. (2016) observed reduced dynamic stability during turns in people with PD, compared to healthy individuals, particularly when patients were asked to perform fast turns. Therefore, the slower turning speeds in people with PD might reflect a compensatory strategy to prevent dynamic postural instability.

3.  I didn't quite understand the rationale that slower turning speeds make it harder to fall.

Isn't it natural for PD patients with FoG to have slow turning speeds?

We found that those with FoG were slower at turning than people with Parkinson’s without FoG, and controls. Furthermore, our results show that visual cueing significantly slowed turn velocity and increased duration, which is the opposite of our hypothesis. As detailed above, this may indicate that people with Parkinson’s took a more cautious approach to turning, which may be safer for them.

Also, when checking the visual stimulus, it is a normal human reaction that the turning speed slows down, and the result is predictable.

Extensive analysis is required using EEG and gaze measurement data to confirm the effect of visual stimuli.

Therefore, I consider the novelty and contribution of this article to be small.

It will be necessary to review the method of hypothesis testing and the method of interpretation.

Although we agree that EEG and gaze outcomes may add valuable information in better understanding the effects of visual cueing, they are beyond the scope of this paper. The effect of the visual cue is not that of a static computer based ‘visual cue’ as used in experimental / behavioural neuroscience research, which aims to elicit eye or brain activity responses (using eye-tracking or EEG). The visual cue used in this study is taped lines on the floor to improve mobility, in this case turning, in Parkinson’s.

This is the first study to examine a visual cue (taped lines on the floor) that is often used in clinical practice to improve gait/turning in Parkinson’s, especially those with freezing of gait. Therefore, we feel there are novelty components to this study. In addition, it provides new data on the turning response in Parkinson’s with the aid of inertial sensors.

This study provides ‘real-world’ clinically relevant data on a commonly used intervention for mobility impairment in Parkinson’s.

Round 2

Reviewer 1 Report

All my concerns were addressed. 

Reviewer 2 Report

The authors have well addressed my concerns, and I can recommend it for a publication.

Reviewer 3 Report

Confirmed the fix.